# Protocol for a randomised feasibility study of Point-Of-care HIV viral load testing to Enhance Re-suppression in South Africa: the POwER study

Jienchi Dorward [1,2] Yukteshwar Sookrajh,[3] Hope Ngobese,[3] Richard Lessells,[2,4] Fathima Sayed,[2] Elliot Bulo,[3] P Moodley,[5,6] Natasha Samsunder,[2] Lara Lewis,[2] Sarah Tonkin-Crine [1,7] Paul K Drain,[8,9,10] Gail Hayward [1] Christopher C Butler [1] Nigel Garrett [2,11]

► Prepublication history and additional material is published online only. To view please visit the journal online (http://dx.doi.org/10.1136/bmjopen-2020-045373).

For numbered affiliations see end of article.

**Correspondence to**
Dr Jienchi Dorward;
jienchi.dorward@phc.ox.ac.uk

## ABSTRACT

**Introduction** Access to HIV viral load testing remains difficult for many people on antiretroviral therapy (ART) in low-income and middle-income countries. Weak laboratory and clinic systems often delay the detection and management of viraemia, which can lead to morbidity, drug resistance and HIV transmission. Point-of-care testing could overcome these challenges. We aim to assess whether it is feasible to conduct a randomised trial of point-of-care viral load testing to manage viraemia.

**Methods and analysis** We will conduct an open-label, single-site, individually randomised, feasibility study of Point-Of-care HIV viral load testing to Enhance Re-suppression, in Durban, South Africa. We will enrol approximately 100 people living with HIV who are aged ≥18 years, receiving first-line ART but with recent viraemia ≥1000 copies/mL, and randomise them 1:1 to receive point-of-care viral load or standard laboratory viral load monitoring, after 12 weeks. All participants will continue to receive care from public sector healthcare workers following South African HIV management guidelines. Participants with persistent viraemia ≥1000 copies/mL will be considered for switching to second-line ART. We will compare the proportion in each study arm who achieve the primary outcome of viral suppression <50 copies/mL at 24 weeks after enrolment. Additional outcomes include proportions retained in the study, proportions with HIV drug resistance, time to viral load results and time to switching to second-line ART. We will assess implementation of point-of-care viral load testing using process evaluation data, and through interviews and focus groups with healthcare workers.

**Ethics and dissemination** University of Oxford Tropical Research Ethics Committee and the Biomedical Research Ethics Committee of the University of KwaZulu-Natal have approved the study. We will present results to stakeholders, and through conferences and open-access, peer-reviewed journals.

**Trial registration number** PACTR202001785886049.

## INTRODUCTION

Initiating antiretroviral therapy (ART) to achieve viral load suppression among all

## Strengths and limitations of this study

► The Point-Of-care HIV viral load testing to Enhance Re-suppression (POwER) study will provide new evidence to guide the development of interventions using point-of-care HIV viral load testing to improve the management of viraemia in low-income and middle-income countries.
► The study will be conducted in a public health facility and will inform the utility and ease of implementation of point-of-care viral load testing in a routine clinical setting.
► The study is limited by a moderate sample size and limited power to detect an effect of point-of-care viral load testing on viral resuppression.

people with HIV is crucial to achieve the Joint United Nations Programme on HIV/AIDS target of ending AIDS by 2030. However, poor adherence to ART and/or HIV drug resistance can lead to HIV viraemia, with associated increases in morbidity, mortality, onward HIV transmission and the development and spread of HIV drug resistance.[1] Viral load testing can identify viraemia and guide adherence counselling, and/or switching to second-line ART regimens if HIV drug resistance is suspected. However, in low-income and middle-income countries (LMICs), the majority of ART is provided in primary care, where there is limited laboratory viral load capacity, weak clinic systems to act on viral load results and a paucity of evidence-based interventions to improve adherence.[1–5] Poor adherence can be due to complex social and psychological issues (eg, gender-based violence, alcohol use, migrant labour and long distances to clinics).[6] Therefore, multiple visits for blood draws and result review confers an additional burden for these

vulnerable populations. Laboratory viral load results are often not reviewed until the next routine clinic visit after several months, by which time adherence problems may have worsened.[1–4]

Point-of-care viral load testing could allow same-day adherence counselling and switching to second-line ART, and more efficient, person-centred care by reducing the burden of clinic visits for review of blood results; thereby leading to faster viral load resuppression. WHO has approved the Xpert HIV-1 VL (Cepheid, Sunnyvale, USA)[7] and the m-PIMA HIV-1/2 VL (Abbott, Chicago, USA)[8] as accurate, quantitative point-of-care HIV viral load assays for use in LMICs. However, these assays have not yet been rolled out for ART monitoring in LMICs, due to concerns around implementation, and with limited randomised trial evidence to support clinical effectiveness. To date there is only one, single-site, randomised trial that has evaluated the clinical effectiveness of these assays. This was the (Simplifying HIV Treatment and Monitoring) STREAM point-of-care viral load trial, conducted by our team, in which 390 adults stable on ART for 6 months were randomised to receive either usual care with laboratory viral load testing, or care by an enrolled nurse and point-of-care viral load testing.[9] After 12 months, people in the intervention arm were more likely to have been referred into a community ART delivery programme, and had 14% higher retention in care and viral suppression.[10] However, the combined intervention was provided by research staff, meaning we do not know if point-of-care viral load assays can be implemented effectively in public sector clinics in LMICs.[11] Furthermore, STREAM only enrolled clinically stable patients (5% had viraemia at enrolment), and so there is inadequate data on the effect of point-of-care viral load testing among people with viraemia who are a vulnerable, priority population.

## OBJECTIVES

The overall aim of this feasibility study[12] is to provide exploratory estimates to guide the development of a future larger trial of point-of-care testing to manage HIV viraemia. Our specific objectives are to:

1. Broadly estimate the effect size of point-of-care viral load testing compared with standard laboratory viral load testing on viral resuppression <50 copies/mL after 24 weeks.
2. Determine the feasibility of recruiting, randomising and following up patients in a randomised trial of point-of-care viral load testing to manage HIV viraemia.
3. Assess the perceptions of staff and the practical changes in clinic systems that are required to implement point-of-care viral load testing for a subset of patients in a primary care clinic in South Africa.

## METHODS AND ANALYSIS
### Trial design

This will be a an open-label, single-site, individually randomised, feasibility study of point-of-care viral load testing for the management of HIV viraemia (figure 1). We aim to

enrol approximately 100 adults receiving first-line ART and who have viraemia, and randomise them in a 1:1 ratio to receive point of care versus standard laboratory-based viral load testing. We will assess viral suppression after 24 weeks in each arm, and estimate recruitment, fidelity of the intervention and retention in care. We will also assess the implementation of point-of-care viral load testing using process evaluation data, in-depth interviews and focus group discussions.

### Study setting

The study will take place at the Prince Cyril Zulu Clinic; a large public sector clinic run by the eThekwini Municipality Health Unit in central Durban, South Africa. The clinic manages approximately 11 000 people receiving ART and follows South African National Department of Health guidelines.[13] Since December 2019, South Africa has started replacing current non-nucleoside reverse transcriptase inhibitors (NNRTIs), with the new drug dolutegravir for first-line ART.[14 15] Dolutegravir is an integrase strand transfer inhibitor that, in high-income settings, has been found to have better tolerability, efficacy and durability than regimens based on NNRTIs such as efavirenz.[14 16] Current first-line ART regimens at Prince Cyril Zulu Clinic are therefore tenofovir disoproxil fumarate, emtricitabine and efavirenz, or tenofovir disoproxil fumarate, lamivudine and dolutegravir.[13 14] Viral load

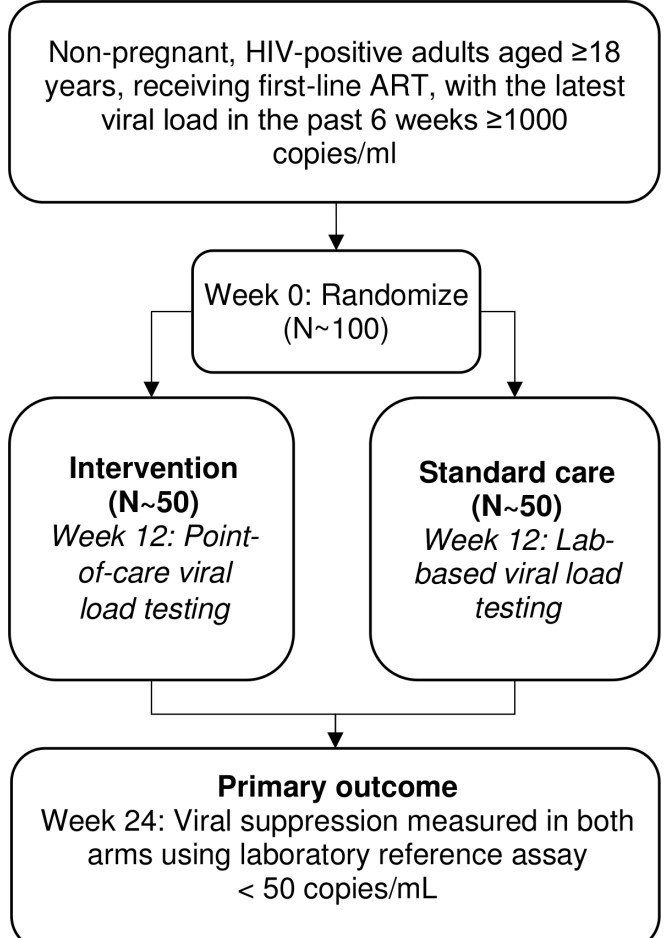

**Figure 1** Consort diagram of the POwER study. ART, antiretroviral therapy.

testing is performed at six and twelve months after ART initiation and annually thereafter. Similar to the WHO guidelines, patients with viraemia are given enhanced adherence counselling, with a repeat viral load within 3 months. If viraemia persists on NNRTI-based regimens, then HIV drug resistance is more likely, and so a switch to second-line ART should be considered.[13 17] For dolutegravir-based regimens, HIV drug resistance is less likely, and so a switch to second-line ART is only recommended if there are signs of clinical or immunological failure, or after 2 years of persistent viraemia.[13]

## Eligibility criteria

Eligible patients will be HIV-positive adults aged ≥18 years, receiving first-line NNRTI-based or dolutegravir-based ART, with the latest viral load in the past 6 weeks ≥1000 copies/mL, and with no previous enhanced adherence counselling for this episode of viraemia. Pregnant women will not be eligible as they are routinely referred out of the Prince Cyril Zulu Clinic for antenatal care.

## Study procedures

### Recruitment

Potential participants will be identified by clinic staff using laboratory reports, electronic clinic information systems and clinic notes.

### Informed consent, screening and enrolment

A research assistant or nurse will describe the study to potential participants, address any questions and seek written informed consent for participation. A research nurse will then assess eligibility using the patient's demographics, and a brief medical history and examination, including pregnancy testing for all women.

### Randomisation and blinding

A statistician will generate the allocation sequence using random numbers generated in SAS V.9.4 (SAS Institute). We will stratify randomisation by first-line ART regimen at enrolment (NNRTI-based or dolutegravir based). The allocation sequence will contain variable block sizes, with participants randomised in a 1:1 ratio to the intervention or standard of care arm. All study staff apart from the statistician and data manager will be blinded to the allocation sequence. The allocation sequence will be programmed into the REDCap V.8.7.1 (Vanderbilt University, Nashville, USA)[18] enrolment electronic case report form (eCRF). At enrolment, the research assistant or research nurse will complete the enrolment eCRF, which will automatically assign the intervention allocation. As this is an open label study, study staff, clinic staff and participants will not be blinded to intervention allocation.

### Baseline assessments

At enrolment, a nurse will administer a baseline sociodemographic questionnaire, and take a clinical history and examination, including an ART history. Throughout the study, South African Department of Health guidelines for managing viraemia will be used.[13] This includes enhanced adherence counselling, which will be performed by clinic staff with participants in both arms using the pre-enrolment viral load result. Techniques used during the counselling include ART education, evaluation of social support, mental health screening and the use of treatment supporters where appropriate.[19] A nurse will also draw blood for routine tests for investigation of viraemia, which currently include CD4 count testing, hepatitis B surface antigen, creatinine and haemoglobin (table 1).[13]

### Follow-Up

Participants in both arms will have routine clinic visits scheduled at the healthcare worker and participant's discretion, which is typically every 28 days to coincide with ART collection. At these visits, clinic staff will provide ongoing enhanced adherence guidelines, clinical support and ART when necessary. South African guidelines recommend a repeat viral load 3 months after the first high result ≥1000 copies/mL. As this study is taking place within a routine care setting, we anticipate that not all participants will attend within the 3-month visit window. In these cases, we will allow repeat viral load to be done at another time, at the attending clinician's discretion.

### Viral load testing interventions

For participants in the point-of-care arm, the repeat viral load test will be performed by a nurse, phlebotomist or laboratory technician in the clinic, using a WHO approved, fully automated point-of-care viral load assay (such as the Xpert HIV-1 VL). Assay run times are ≤90 min. Clinic staff will be encouraged to provide results in the same clinic visit to inform ongoing management. If the participant cannot wait, or results are not available in the same visit, they will be available at the next clinic visit scheduled at staff and participant's discretion.

In the standard of care arm, samples for viral load testing will be transported on the same day to the National Health Laboratory Service (NHLS) for testing using their routine laboratory assays (eg, Alinity m HIV-1 assay (Abbott, Chicago, USA)). Results are normally available after 7 days and will be provided to the participant at the next clinic visit, arranged at the participants and healthcare worker's discretion (typically after 7–28 days for results, depending on the participant's availability, ART supply and clinic schedules).

### Management of viral load results at routine follow-up clinic visits

Viral load results will be managed by clinic staff in accordance with South African guidelines, which include guidance for management of both efavirenz and dolutegravir-based regimens, and do not advise routine HIV drug resistance testing.[20] Participants with a viral load ≥1000 copies/mL may meet criteria for viral failure (two consecutive viral loads ≥1000 copies/mL over 3 months apart) and will be considered for a switch to second-line ART. Switching to second-line ART and the choice of regimen will be at the clinician's discretion and will depend on the participant's first-line ART regimen, previous ART history, participant's preferences, perceived

**Table 1** Schedule of evaluation in the POwER study

| Weeks in study | Enrol 0 | Routine clinic visit follow-up* 4 | 8 | 12 | 16 | 20 | Study exit 24† |
|---|---|---|---|---|---|---|---|
| Informed consent | X | | | | | | |
| Locator information | X | | | | | | |
| Demographics | X | | | | | | |
| Medical history | X | | | | | | |
| Vital signs | X | X | X | X | X | X | X |
| Physical examination | X | | | | | | |
| Urine pregnancy test | X | | | | | | |
| Eligibility screen | X | | | | | | |
| Enhanced adherence counselling | X | X | X | X | X‡ | X‡ | X‡ |
| Randomisation | X | | | | | | |
| Routine Department of Health testing§ | X | | | | | | |
| POC viral load (intervention arm) | | | | X¶ | | | |
| Laboratory viral load (standard of care arm) | | | | X¶ | | | |
| Reference viral load outcome measure | | | | | | | X |
| Evaluation of new POC assays** | X | | | X | | | X |
| Stored blood and HIV drug resistance testing†† | X | | | X | | | X |
| Stored urine for ART drug level testing | X | | | | | | |

*Scheduled at participants and healthcare worker's discretion but typically every 28 days.
†Study exit visit window 22–30 weeks.
‡Ongoing enhanced adherence counselling for participants with viral load ≥50 copies/mL.
§CD4 count testing, hepatitis B surface antigen, creatinine and haemoglobin.
¶Repeat viral load testing recommended at 12 weeks but may be done any time during follow-up at the attending clinician's discretion.
**Maximum of 8 mL venous blood and/or a finger-prick capillary blood sample.
††Retrospective drug resistance testing for viraemic participants at enrolment, follow-up or study exit (10 mL EDTA).
ART, antiretroviral therapy; POC, point-of-care; POwER, Point-Of-care HIV viral load testing to Enhance Re-suppression.

adherence and current guidelines.[20] Participants with a second viral load <1000 copies/mL will remain on first-line ART. In accordance with guidelines for the transition to dolutegravir, those receiving an NNRTI may have this switched to a dolutegravir-based first-line regimen.[13]

### Data capture during routine clinic visits

Research staff will monitor participants' clinical records to capture visit dates, whether enhanced adherence counselling was conducted by a counsellor, nurse or doctor, medication prescriptions including ART and laboratory results.

### Study exit visit

The study exit visit will occur at 24 weeks (visit window 22–30 weeks), after which participants will continue receiving standard care at the Prince Cyril Zulu Clinic. At 24 weeks, any participant who has not attended the study exit visit will be called by the research team and asked to attend, irrespective of whether they have a routine clinic visit scheduled in the exit visit window.

### Participant reimbursement

We will reimburse study participants ZAR150 (approximately £8.00) after the enrolment visit and ZAR100

(£5.50) after the 24-week visit, to cover any additional time spent at the clinic and any inconvenience due to study participation.[21]

### Extended follow-up of routine data

For participants who provide additional consent, research staff may access routine medical records for up to 5 years after the study exit visit, in order to determine longer term retention in care, ART adherence and viral load results. For participants who are lost to follow-up, and who provided consent at enrolment, the participants mortality status may be checked on the South African National Population Register.

### Additional laboratory specimens

Participants who provide additional consent will also have 10 mL of venous blood taken for storage and retrospective ART drug level and HIV drug resistance testing. Plasma will be stored at enrolment, during follow-up alongside the repeat viral load test, and at study exit. In addition, at enrolment, 5 mL of urine will be taken for storage and ART drug-level testing. We may use study samples to validate new point-of-care viral load, HIV drug resistance or

**Table 2** Study objectives and outcomes in the power study

| Objectives | Outcome measures | Time point(s) of evaluation of this outcome measure (if applicable) |
|---|---|---|
| **Primary objective**<br>To estimate the effect of implementing point-of-care viral load testing versus standard laboratory viral load testing on the proportion of HIV positive participants with viraemia who subsequently achieve viral resuppression | The proportion of participants in each arm with viral load <50 copies/mL measured on laboratory reference assay | 24 weeks after enrolment |
| **Secondary objectives**<br>To assess whether it is feasible to perform a randomised implementation trial of point-of-care viral load testing to manage viraemia in a routine South African clinic, by determining: | | |
| (A) What proportion of people with HIV viraemia is feasible to enrol? | (A) Proportion of patients with viraemia at the study clinic who are successfully enrolled in the study | (A) During study enrolment period |
| (B) What proportion of those enrolled is feasible to follow-up? | (B) Proportion of enrolled participants who attend the study exit visit overall and in each arm | (B) 24 weeks after enrolment |
| (C) What proportion of point-of-care viral load results are received and acted on in the same day? | (C1) The proportion of point-of-care viral load tests that are communicated to participants on the same day and (C2) The proportion of point-of-care viral load tests ≥1000 copies/mL that result in same day enhanced adherence counselling and (C3) The proportion of point-of-care viral load tests ≥1000 copies/mL that result in same day switch to second-line ART | (C) 12 weeks after enrolment when point-of-care testing is performed |
| **Tertiary Objectives**<br>To estimate the effect of point-of-care viral load testing versus standard laboratory viral load testing on: | | |
| (A) Time to detection of viral failure (consecutive viral loads ≥1000 copies/mL) in each arm | (A) Days from enrolment to availability of viral load result ≥1000 copies/mL | (A) By 24 weeks after enrolment |
| (B) Time to switch to second-line ART in each arm | (B) Days from enrolment to appropriate switching to second-line ART, among participants with viral failure | (B2) By 24 weeks after enrolment |
| (C) Appropriate switching to dolutegravir in each arm | (C) Time to appropriate switching to dolutegravir in each arm | (C3) By 24 weeks after enrolment |
| (D) HIV drug resistance in each arm | (D) The proportion of participants in each arm with HIV drug resistance | (D4) 24 weeks after enrolment |
| **Qualitative study objectives:**<br>What changes in clinic systems are required and what are the views and experiences of staff in implementing point-of-care viral load testing to manage viraemia? | Staff perspectives regarding implementation of point-of-care viral load testing in a routine clinic | During enrolment when point-of-care testing is being implemented, and after study conclusion |

ART, antiretroviral therapy.

ART drug level assays. These results would not be used to guide clinical management.

### Study outcomes

We aim to broadly estimate the effect of point-of-care viral load testing using the primary outcome of viral suppression <50 copies/mL at 24 weeks after enrolment, measured on a laboratory reference assay (eg, Alinity m HIV-1) in both arms. Feasibility outcomes include proportions of participants enrolled, followed up and

who had same-day viral load testing in the point-of-care arm (table 2).

### Data collection and management

The research team will capture study data using standardised electronic CRFs in REDCap, which is a password protected data management service stored on a secure server.[18] All data entry will undergo three stages of quality control including preprogrammed data validity checks in electronic CRFs, immediate source document review

and weekly quality reports generated using REDCap. To protect confidentiality, all study-specific documents, other than the signed consent, will refer to the participant using their study participant number rather than their name. All documents will be stored in a secure cabinet in the locked research office and will only be accessible by study staff and authorised personnel.

### Statistical methods

At the conclusion of the study, we will assess the proportion of participants achieving study outcomes in each arm, with 95% CIs calculated using the Wilson method. We will conduct exploratory analyses using Fisher's exact test to compare the proportions achieving binary study outcome measures in each arm, using an intention to treat analysis. As this is a feasibility study, these exploratory analyses will likely not be powered to test the hypotheses that there is a difference between the two arms for most outcome measures.

### Primary outcome

We will calculate the proportion of participants in each arm achieving the binary primary outcome of viral suppression <50 copies/mL at 24 weeks (table 2). Participants who are lost to follow-up with no viral load result will be included in the analysis as not having achieved the primary outcome.

### Secondary outcomes

We will calculate the proportion of potentially eligible patients with viraemia at the Prince Cyril Zulu clinic who are enrolled in the study (table 3). The denominator will be the number of patients with viraemia during the study enrolment period, which will be calculated by clinic staff using NHLS high viral load reports. These reports are routinely used for monitoring and evaluation of clinic performance and include the number of first high viral loads ≥1000 copies/mL taken from the Prince Cyril Zulu Clinic each week. To assess study follow-up, we will calculate the proportion of participants who attend the study exit visit at 24 weeks after enrolment, overall and in each arm in case point-of-care testing influences retention. To assess implementation of point-of-care viral load testing, we will calculate the proportion of point-of-care viral load tests which are provided to participants on the same day, the proportion of viral loads ≥1000 copies/mL which resulted in same day enhanced adherence counselling, and the proportion of viral loads ≥1000 copies/mL resulting in appropriate same-day switch to second-line ART.

### Tertiary outcomes

In each arm, we will assess the median number of days (and IQR) from enrolment to detection of viral failure, and from enrolment to appropriate switch to second-line ART (among participants with viral failure), and from enrolment to appropriate switching to dolutegravir (among participants on an NNRTI-based regimen). We will compare time to event outcomes using Cox proportional hazards. We will also calculate the proportion of participants with HIV drug resistance detected at 24 weeks after enrolment.

### Sample size considerations

We conservatively estimate (based on our previous study)[9 10] that we could enrol approximately 100 participants in 6 months. If enrolment proceeds better than anticipated, and resources allow us to enrol more, this will provide more precise estimates of the study outcomes. If enrolment proceeds slower than expected, we will consider expanding to another clinic site to achieve a minimum of 80 participants. Therefore, while we anticipate 100 participants, the final number may be between approximately 80–180 participants, and will be determined by the time and resources available. We will use NHLS, Prince Cyril Zulu Clinic and study data to assess outcomes, with estimated precision, in table 3. Although not the primary aim of this feasibility study, assuming 50% of standard care participants achieve the main outcome of viral load suppression <50 copies/mL at 24 weeks, a sample size of 100 participants (50 per study arm) would provide 91% power to detect a +30% difference in the point-of-care testing arm, using a two-sided alpha of 0.05.

### Quality assurance procedures

Study documentation will be subject to internal quality audits by the Centre for the AIDS Programme of Research in South Africa (CAPRISA) Quality Assurance team, in accordance with CAPRISA standard operating procedures.

**Table 3** Study and process evaluation outcomes with precision estimates

| Outcome | Estimated n/N, % | 95% CI (Wilson) |
| --- | --- | --- |
| Percentage with viral load <50 copies/mL at 24 weeks in POC arm | 35/50, 70.0 | 56.2 to 80.9 |
| Percentage with viral load <50 copies/mL at 24 weeks in SOC arm | 25/50, 50.0 | 36.6 to 63.4 |
| Percentage of viraemic patients successfully enrolled in study | 100/240*, 41.7 | 35.6 to 48.0 |
| Percentage of those enrolled who are retained at 24 weeks | 90/100, 90.0 | 82.6 to 94.5 |
| Percentage in POC arm with same-day viral load testing | 40/50, 79.5 | 70.0 to 88.8 |

*Assuming 240 viraemic patients during enrolment period, many of whom will not be eligible due to being on second-line ART, having had previous high viral loads, already having received enhanced adherence counselling or pregnancy.
ART, antiretroviral therapy; POC, point-of-care; SOC, standard-of-care.

## Qualitative substudy

We will undertake a qualitative substudy within Point-Of-care HIV viral load testing to Enhance Re-suppression (POwER) to assess what changes in clinic systems are required and what the views of staff are in implementing point-of-care viral load testing, its potential impact on adherence and adherence counselling and managing viraemia. We will conduct semistructured interviews and a focus group discussion with approximately 8–10 staff during, and then again after, implementation of point-of-care viral load testing in the study. Staff will include counsellors, phlebotomists, nurses, pharmacy staff, laboratory staff, doctors and health service managers. Topic guides will be informed by normalisation process theory which aims to identify what is needed to 'normalise' use of a technology in a healthcare system.[22][23] Discussions will be transcribed and thematically analysed using NVIVO software (QSR International, Melbourne, Australia).

## Patient and public involvement

Patients were not involved in the design of this study.

## Ethics and dissemination

We will conduct the trial in accordance with the principles of the Declaration of Helsinki. The eThekwini Municipality Health Unit Research Committee, the KwaZulu-Natal Provincial Health Research Ethics Committee (KZ_202002_005), the University of KwaZulu-Natal Biomedical Research Ethics Committee (BREC/00000836/2019) and the University of Oxford Tropical Research Ethics Committee (OxTREC 66–19) have approved the study. We will present study results to front-line clinical workers and programme managers in eThekwini Municipality and KwaZulu-Natal Provincial Department of Health. We will also submit our findings to academic conferences and open-access, peer-reviewed journals.

## DISCUSSION

Evidence-based interventions to improve the management of viraemia are needed to achieve global HIV treatment targets, prevent the development and spread of HIV drug resistance, and improve patient outcomes. In this paper, we outline a feasibility trial of point-of-care viral load testing among people with viraemia who are receiving first-line ART in South Africa. Results from this study will inform the design of future trials to assess the clinical effectiveness of point-of-care testing to manage viraemia.

### Problems with management of viraemia

Several studies highlight gaps in the management of viraemia in LMICs, and the resulting negative impacts on treatment outcomes. First, addressing potential poor adherence, which may have caused viraemia, is challenging. A meta-analysis of interventions to improve adherence did not find any strategies that resulted in better viral suppression in LMICs,[24] while a more recent cluster randomised trial of the South African National Adherence Guidelines found no difference in long-term viral resuppression among patients with viraemia.[5] Second, follow-up viral load testing is often poorly performed. In the South African trial, less than 20% of patients had a repeat viral load within the recommended 3 months to confirm viral failure.[5] Similar delays have been reported in a large analysis of South African laboratory data; among 260 323 patients with viraemia, the median time to a repeat viral load was 6.2 months, (IQR 4.0–11.1)[25]. Third, among people with confirmed viral failure, a systematic review,[26] and more recent individual studies,[4][27–29] have found that only 30%–60% are switched to second-line ART. Even among patients who are switched, this can take a further 3–12 months,[27–29] with delays of even a few months being associated with increased risk of opportunistic infections and mortality, particularly among people with low CD4 counts.[1][4][27–29] Taken together, these data highlight the need for interventions to reduce delays and improve the management of viraemia.

### Existing research regarding clinical effectiveness of point-of-care viral load testing

We conducted the STREAM trial, which was the first randomised trial of point-of-care HIV viral load testing for monitoring ART.[9][10] Overall, the STREAM intervention improved retention in care and viral suppression by 14.3% compared with the standard-of-care laboratory viral load arm. However, STREAM was designed to enrol stable patients; less than 5% of participants had viraemia at enrolment, and only 16 participants developed viral failure (seven in the point-of-care arm, nine in the standard care arm). Time to detection of viral failure was faster in the intervention arm (55 days (IQR 55–57) vs 123 days (IQR 98–162)) but numbers with viraemia were too small to determine whether point-of-care testing was effective at improving switching to second line ART, or subsequent viral load outcomes. In STREAM, the research team provided point-of-care testing and clinical care, so the observed benefits may also not be applicable to routine public sector settings.

Several more trials of point-of-care viral load testing are in progress, but none focus on management of viraemia. Studies in Zimbabwe,[30][31] Kenya[32] and Haiti,[33] are assessing point-of-care viral load for monitoring ART among pregnant women, children and/or adolescents. A study in Nigeria is enrolling adults at ART initiation and will report on the effect of point-of-care viral load testing after 12 months,[34] while a trial in South Africa[35] is enrolling adults on ART who are due for their annual viral load test. One trial in Uganda is assessing a viral load intervention package that includes 'near point-of-care testing' (at the clinic or a nearby clinic and therefore results may not be

provided on the same day), and is enrolling 'high-risk groups' including patients with previous viraemia or no viral load in the past year.[36]

## Implementation of point-of-care viral load testing

Point-of-care diagnostics have been used widely in healthcare systems in many LMICs.[37] In HIV programmes, some tests such as rapid, lateral flow assays for diagnosis of HIV, have been evaluated, endorsed and incorporated into WHO guidelines and successfully adopted in many settings.[38] However, other assays, such as more complex molecular PCR technologies for tuberculosis, have remained as laboratory tests despite being marketed as point-of-care assays.[37] Given that the Xpert HIV-1 VL and m-PIMA assays use similar platforms and take over 1 hour for results to be available, they have also been criticised as not being implementable as true point-of-care tests. A large systematic review of barriers to point-of-care HIV diagnostic implementation in LMICs found that in 132 studies, integration of the point-of-care test into clinical work flows was the most commonly identified challenge to test utilisation.[39] More recently, studies have reported implementation of qualitative point-of-care viral load assays for early infant diagnosis of HIV,[40] and monitoring of ART in maternity wards in South Africa,[41] and decentralised ART clinics in Malawi.[42] While these studies have reported quantitative outcomes related to point-of-care viral load use, there is little qualitative work using theory-based approaches to evaluate what is required for successful implementation.

## Strengths and limitations

A major strength of this study is the focus on patients with viraemia, who are a priority population at high risk of increased morbidity and mortality, onwards HIV transmission and development of HIV drug resistance.[1] Interventions to improve viral suppression among this group have been prioritised by the WHO and align with several objectives in the Global Action Plan on HIV drug resistance.[43] While our study may be underpowered to detect a significant effect of point-of-care viral load testing on viral resuppression, we will determine the feasibility of conducting larger studies to answer this question in this setting. Our use of normalisation process theory will allow a rigorous assessment of what is needed to 'normalise' point-of-care testing in the South African healthcare system.

A limitation of this study is the focus on 'supply-side' implementation of point-of-care viral load testing. Receiving results on the same day could also increase patient understanding and motivate adherence, but evaluating this from the patient perspective is beyond the scope of the study. Furthermore, findings from this large urban, clinic may not be applicable to other settings, including smaller clinics in rural settings where advantages of point-of-care testing over laboratory-based testing may differ. Our future trial may be best designed as a cluster randomised trial, with individual clinics randomised to receive the point-of-care testing intervention. While POwER is an individually randomised study, estimates of the effect of point-of-care testing will remain valuable for design of potential cluster randomised or individually randomised future studies.

Coinciding with the rollout of dolutegravir is both a strength and a weakness of our study. We will enrol patients on either NNRTI- or dolutegravir-based ART, therefore, allowing estimates of the effect of point-of-care testing in either group. For patients enrolled on NNRTIs, this also presents an opportunity to assess whether point-of-care viral load testing can assist the transition to dolutegravir-based regimens. For patients enrolled on dolutegravir, a potential limitation is that point-of-care viral load testing will not be used to expedite a switch to second line, although it may still improve enhanced adherence counselling and allow more efficient care that can help ensure retention and virological suppression. Lastly, we will be able to assess early viral suppression outcomes and the frequency of HIV drug resistance among people with viraemia on dolutegravir. This is a pressing research question for global policy makers, because it determines the need for HIV drug-resistance testing and second-line ART after introduction of a dolutegravir-based first-line in LMICs.

In summary, we aim to provide evidence as to whether a trial of point-of-care viral load testing to manage viraemia is feasible, and how testing may be implemented in South African clinics. This work will contribute to the development of interventions to improve the management of viraemia among people receiving ART, thereby improving health and leading to longer and better lives, as well as reducing the spread of HIV and preventing drug resistance.

**Author affiliations**
[1]Nuffield Department of Primary Care Health Sciences, University of Oxford, Oxford, UK
[2]Centre for the Aids Programme of Research in South Africa, University of KwaZulu-Natal, Durban, South Africa
[3]eThekwini Municipality Health Unit, Durban, South Africa
[4]KwaZulu-Natal Research and Innovation Sequencing Platform (KRISP), University of KwaZulu-Natal, Durban, South Africa
[5]Department of Virology, University of KwaZulu-Natal, Durban, South Africa
[6]National Health Laboratory Service, Inkosi Albert Luthuli Central Hospital, Durban, South Africa
[7]NIHR Health Protection Research Unit in Healthcare Associated Infections and Antimicrobial Resistance, University of Oxford, Oxford, UK
[8]Department of Global Health, Schools of Medicine and Public Health, University of Washington, Seattle, Washington, USA
[9]Department of Medicine, School of Medicine, University of Washington, Seattle, Washington, USA
[10]Department of Epidemiology, School of Public Health, University of Washington, Seattle, Washington, USA
[11]Discipline of Public Health Medicine, School of Nursing and Public Health, University of KwaZulu-Natal, Durban, South Africa

**Acknowledgements** The authors would like to thank all participants in the study and acknowledge the work and support of staff at the Prince Cyril Zulu Clinic, eThekwini Municipality, CAPRISA and the National Health Laboratory Services at Addington and Inkosi Albert Luthuli Hospitals.

**Contributors** JD and NG conceived the study. YS, HN, RL, FS, EB, PM, NS, LL, ST-C, PKD, GH and CCB contributed to study design and implementation. ST-C contributed to design of the qualitative substudy. JD wrote the first draft of the manuscript. All authors critically reviewed and edited the manuscript and consented to final publication.

**Funding** This work is supported by grants from the Wellcome Trust PhD Programme for Primary Care Clinicians (216421/Z/19/Z), the University of Oxford's Research England QR Global Challenges Research Fund (0007365) and the Africa Oxford Initiative (AfiOx-119). The University of Oxford is the study sponsor.

**Disclaimer** The funders and sponsor have no role in study design, manuscript submission, or collection, management, analysis or interpretation of study data.

**Competing interests** Cepheid and Abbott have provided point-of-care viral load assays at no cost for use at the study site.

**Patient consent for publication** Not required.

**Provenance and peer review** Not commissioned; externally peer reviewed.

**ORCID iDs**
Jienchi Dorward http://orcid.org/0000-0001-6072-1430
Sarah Tonkin-Crine http://orcid.org/0000-0003-4470-1151
Gail Hayward http://orcid.org/0000-0003-0852-627X
Christopher C Butler http://orcid.org/0000-0002-0102-3453
Nigel Garrett http://orcid.org/0000-0002-4530-234X

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
