## [Reviewer comments · BMJ Open]

ARTICLE DETAILS

TITLE (PROVISIONAL)	Protocol for a randomised feasibility study of Point-Of-care HIV viral load testing to Enhance Re-suppression in South Africa: The POWER study
AUTHORS	Dorward, Jienchi; Sookrajh, Yuktेशwar; Ngobese, Hope; Lessells, Richard; Sayed, Fathima; Bulo, Elliot; Moodley, P; Samsunder, Natasha; Lewis, Lara; Tonkin-Crine, Sarah; Drain, Paul; Hayward, Gail; Butler, Christopher C.; Garrett, Nigel

VERSION 1 – REVIEW

REVIEWER	Nancy Puttkammer University of Washington, Seattle WA, USA One of the authors, Paul K Drain, is a colleague within the UW Department of Global Health. However, I have not collaborated directly with Dr. Drain, and I do not know any of the other co-authors.
REVIEW RETURNED	02-Dec-2020

GENERAL COMMENTS	This is a well-written protocol paper which clearly explains the present study, the precursor work and why the research will contribute useful evidence to the field. I have several suggestions to provide further clarity to the reader. Methods  • Random allocation. Do the authors expect that allocation will be balanced by time on ART? Did you consider block randomization stratified by time on ART? Time on ART could be an important factor affecting the outcome of interest, where potential imbalance in allocation could bias the study results. Are there any other factors which could be imbalanced and which could affect the outcome? Are there plans in the analysis stage to adjust for any imbalanced factors, e.g. in sensitivity analyses? • The hypothesized mechanism of action could be more clearly presented in lines 89-91. The authors seem to hypothesize that the intervention effect will be driven by closer adherence to clinical guidelines, and more timely follow up (health services supply side factors). The authors do not posit that the intervention would directly affect patients' knowledge, motivation, self-efficacy for treatment adherence, or other patient-side factors (though these factors could be affected through the enhanced adherence counseling). I believe that there is some case to be made that real-time feedback to patients on their viral suppression status could directly affect adherence motivation (above and beyond the provision of enhanced adherence counseling). It sounds like the researchers do not propose to directly measure adherence motivation or even ART adherence. If this is the case, it could be
--

	good to acknowledge that such a fuller exploration of the mechanisms of action is outside of the scope of their research.  • The 24-week follow up outcome measurement could be framed more clearly. I believe that the authors propose to do a third viral load test at 24 weeks, using a laboratory reference assay (lines 229-30). I wonder if it can be more clearly shown in the Consort figure when the 3 VL tests take place, and that the outcome measure is a study-specific third VL test to be administered to all participants at 24 weeks (both arms). Since the authors indicate that the follow up visits for clients in both arms will be per normal clinic procedures (monthly or at a frequency based on clinician and patient discretion), it will be helpful to clarify how they plan to handle the timing of the 24-week outcome measure. Will this be done during routine visits, or an “off cycle” study specific visit? • Measurement of enhanced adherence counseling (secondary outcome). It will be helpful to understand how the provision of this service is being documented, during both the 0-12 week period and also during the 12-24 week follow up period. This is critical to the implied mechanism of action for the intervention, so it seems important to sensitively and specifically measure the type and intensity of adherence counseling which was actually provided in both arms. Further detail for the reader on how adherence counseling is documented could be helpful. • On a minor point, I notice a more up-to-date citation for reference #31: Reif LK, Belizaire ME, Seo G, et al. Point-of-care viral load testing among adolescents and youth living with HIV in Haiti: a protocol for a randomised trial to evaluate implementation and effect. BMJ Open. 2020;10(8):e036147. Published 2020 Aug 31. doi:10.1136/bmjopen-2019-036147
--	--

REVIEWER	Judith Hahn University of California, San Francisco, CA
REVIEW RETURNED	03-Dec-2020

GENERAL COMMENTS	The trial manuscript is well-written. It describes a trial of routine versus rapid HIV viral load testing. The goal is to increase feedback of the potential for treatment failure that could be accelerated by rapid or point of care HIV testing. The study has been registered with the Pan African Clinical Trials Registry and is currently recruiting. I have several concerns about the study design and some important things are missing from the manuscript. Some can be addressed by better descriptions in the methods. Those facets that cannot be changed at this point need to be clearly noted in the discussion. Introduction: The pros and cons and of the 2 testing strategies are not presented. It is not clear at all whether there is equipoise, in terms of testing cost, testing validity (sensitivity and specificity), ease of administration, etc. If the Xpert testing is clearly advantageous on 2 fronts, there is really no need for a trial, its ethics may be questions, and care should uptake the new testing as soon as is possible. Methods: The trial is presented as a feasibility study (line 116), yet the primary outcome is viral suppression at 24 weeks, which is more of an efficacy endpoint. Feasibility endpoints would be accrual and retention, etc. Because power calculations are based on viral suppression, I take this to be the primary endpoint, so this could perhaps be a pilot trial, but it is not a feasibility trial. The
---

	feasibility outcomes appear to be secondary outcomes. In addition, the aims of the qualitative study , to assess the feasibility of implementation within clinics will be hard to assess because there is no clinic-wide implementation as part of this trial, but random assignment within the clinic. The expected dates and rate of recruitment are not presented, nor a sample size in the methods, until lines 274-275 in which a period of 6 months is mentioned. This is a large omission and a highly non-standard way to conduct a RCT – usually the sample size is pre-determined, and back up plans are made if for some reason recruitment is lower than expected. Table 3 notes that 240 might be eligible, and 100 are anticipated to be enrolled. Why so low? Also this should be discussed in the manuscript text, not just the table. It would be a severe study limitation. The number 240 should also be included in the figure. The motivation for the feasibility outcomes to inform a future larger trial is questionable, given that larger trials (if really needed or ethical, see above), should be conducted as cluster RCTs so that clinic level barriers/facilitators, which are may drive efficacy more than patient variability, can be examined. There is no justification for feasibility outcomes (a and b) for this type of trial – many trials have been conducted in public clinics and the extra anticipated barriers for this particular trial are not described. The outcomes of the point of care testing (i.e. same day results) are justified. Discussion: The discussion does not address some serious limitations of the study. There is no discussion of whether a 90-minute wait is really point of care. There is no discussion of the limitations of a single site trial. There is no discussion of the ramifications of the site chosen for the trial -- a large clinic – and whether it would be representative of a large sector of care. There is no discussion of the limitations of individual randomization versus a cluster trial design. The study design with uncertain sample size is a huge limitation, and there is no discussion of this. The justification that this is a feasibility study does not make sense – even feasibility studies have set sample sizes, and as noted above, the primary outcome is not a feasibility outcome.
--	--

VERSION 1 – AUTHOR RESPONSE

Reviewer: 1

Comment 1: This is a well-written protocol paper which clearly explains the present study, the precursor work and why the research will contribute useful evidence to the field. I have several suggestions to provide further clarity to the reader.

Response: We thank the reviewer for their review and comments.

Comment 2: Methods: Random allocation. Do the authors expect that allocation will be balanced by time on ART? Did you consider block randomization stratified by time on ART? Time on ART could be an important factor affecting the outcome of interest, where potential imbalance in allocation could bias the study results. Are there any other factors which could be imbalanced and which could affect the outcome? Are there plans in the analysis stage to adjust for any imbalanced factors, e.g. in sensitivity analyses?

Response: Thank you for this comment. We do not anticipate that our 1:1 randomisation would be imbalanced by time on ART. We felt that the main factor that would affect the outcome of interest is being on an EFV versus DTG first-line regimen, and therefore stratified randomisation on this variable. We did not feel that any other factors were associated strongly enough with the outcome to warrant further stratification, particularly given the small sample size. Similarly, given the small sample size in these exploratory analyses, we do not plan to include covariates in the outcome analysis.

Comment 3: The hypothesized mechanism of action could be more clearly presented in lines 89-91. The authors seem to hypothesize that the intervention effect will be driven by closer adherence to clinical guidelines, and more timely follow up (health services supply side factors). The authors do not posit that the intervention would directly affect patients' knowledge, motivation, self-efficacy for treatment adherence, or other patient-side factors (though these factors could be affected through the enhanced adherence counselling). I believe that there is some case to be made that real-time feedback to patients on their viral suppression status could directly affect adherence motivation (above and beyond the provision of enhanced adherence counselling). It sounds like the researchers do not propose to directly measure adherence motivation or even ART adherence. If this is the case, it could be good to acknowledge that such a fuller exploration of the mechanisms of action is outside of the scope of their research.

Response: Thank you for this comment. We agree that same day viral load feedback could improve ART adherence. We have added that we will be able to explore this from a staff perspective in our qualitative work, but acknowledge that assessing this from the patient perspective is beyond the scope of this study in the discussion.

Line 318: "We will undertake a qualitative sub-study within POWER to assess what changes in clinic systems are required and what the views of staff are in implementing point-of-care viral load testing, its potential impact on adherence and adherence counselling, and managing viraemia"

Line 416: "A limitation of this study is the focus on 'supply side' implementation of point-of-care viral load testing. Receiving results on the same day could also increase patient understanding and motivate adherence, but evaluating this from the patient perspective is beyond the scope of the study."

Comment 4: The 24-week follow up outcome measurement could be framed more clearly. I believe that the authors propose to do a third viral load test at 24 weeks, using a laboratory reference assay (lines 229-30). I wonder if it can be more clearly shown in the Consort figure when the 3 VL tests take place, and that the outcome measure is a study-specific third VL test to be administered to all participants at 24 weeks (both arms). Since the authors indicate that the follow up visits for clients in both arms will be per normal clinic procedures (monthly or at a frequency based on clinician and patient discretion), it will be helpful to clarify how they plan to handle the timing of the 24-week outcome measure. Will this be done during routine visits, or an "off cycle" study specific visit?

Response: Thank you for this suggestion. We have made these changes to the consort diagram, and outline study exit visit scheduling below:

Line 226: “At 24 weeks, any participant who has not attended the study exit visit will be called by the research team and asked to attend, irrespective of whether they have a routine clinic visit scheduled in the exit visit window.”

Comment 5: Measurement of enhanced adherence counselling (secondary outcome). It will be helpful to understand how the provision of this service is being documented, during both the 0-12 week period and also during the 12-24 week follow up period. This is critical to the implied mechanism of action for the intervention, so it seems important to sensitively and specifically measure the type and intensity of adherence counselling which was actually provided in both arms. Further detail for the reader on how adherence counselling is documented could be helpful.

Response: Thank you for this comment. In the clinic, enhanced adherence counselling is provided in accordance with South African guidelines by non-research staff, and is not specifically part of the intervention, although we agree that it forms part of the mechanism of action. We do not record specific details of what was covered in each counselling session, as this could in itself influence the type of counselling given. In the manuscript we explain what enhanced adherence counselling generally consists of, how it will be captured, and how we will explore the impact of the intervention on adherence counselling in the qualitative work with clinic staff:

Line 181: “This includes enhanced adherence counselling, which will be performed by clinic staff with participants in both arms using the pre-enrolment viral load result. Techniques used during the counselling include ART education, evaluation of social support, mental health screening and the use of treatment supporters where appropriate.¹⁹”

Line 221: “Research staff will monitor participants’ clinical records to capture visit dates, whether enhanced adherence counselling was conducted by a counsellor, nurse or doctor, medication prescriptions including ART, and laboratory results.”

Line 318: “We will undertake a qualitative sub-study within P^OWER to assess what changes in clinic systems are required and what the views of staff are in implementing point-of-care viral load testing, its potential impact on adherence and adherence counselling, and managing viraemia.”

Comment 6: On a minor point, I notice a more up-to-date citation for reference #31: Reif LK, Belizaire ME, Seo G, et al. Point-of-care viral load testing among adolescents and youth living with HIV in Haiti: a protocol for a randomised trial to evaluate implementation and effect. *BMJ Open*. 2020;10(8):e036147. Published 2020 Aug 31. doi:10.1136/bmjopen-2019-036147

Response: We have updated this reference as suggested: 31. Reif LK, Belizaire ME, Seo G, et al. Point-of-care viral load testing among adolescents and youth living with HIV in Haiti: a protocol for a randomised trial to evaluate implementation and effect. *BMJ Open* 2020;10(8):e036147. doi: 10.1136/bmjopen-2019-036147

Reviewer: 2

Comment 1: The trial manuscript is well-written. It describes a trial of routine versus rapid HIV viral load testing. The goal is to increase feedback of the potential for treatment failure that could be accelerated by rapid or point of care HIV testing. The study has been registered with the Pan African Clinical Trials Registry and is currently recruiting. I have several concerns about the study design and some important things are missing from the manuscript. Some can be addressed by better descriptions in the methods. Those facets that cannot be changed at this point need to be clearly noted in the discussion.

Response: Thank you for your review and important comments which we feel add rigour and improve the manuscript.

Comment 2: Introduction: The pros and cons and of the 2 testing strategies are not presented. It is not clear at all whether there is equipoise, in terms of testing cost, testing validity (sensitivity and specificity), ease of administration, etc. If the Xpert testing is clearly advantageous on 2 fronts, there is really no need for a trial, its ethics may be questions, and care should uptake the new testing as soon as is possible.

Response: Thank you for this comment. We do not believe that clinical equipoise is lacking – to date there is only one (single site) randomised trial demonstrating a clinical effect of point-of-care testing, which we describe below. Further evidence in different clinical populations (i.e. viraemic patients), and in larger studies, is needed. We also highlight below some of the reasons why point-of-care viral load testing is not currently used as standard care in low- and middle-income countries, and the gaps in the evidence base which this study addresses:

Line 92: “The World Health Organization (WHO) has approved the Xpert HIV-1 VL (Cepheid, Sunnyvale, USA)⁷ and the m-PIMA HIV-1/2 VL (Abbott, Chicago, USA)⁸ as accurate, quantitative point-of-care HIV viral load assays for use in LMICs. However, these assays have not yet been rolled out for ART monitoring in low- and middle-income countries, due to concerns around implementation, and with limited randomised trial evidence to support clinical effectiveness. To date there is only one, single-site, randomised trial that has evaluated the clinical effectiveness of these assays. This was the STREAM point-of-care viral load trial, conducted by our team, in which 390 adults stable on ART for six months were randomised to receive either usual care with laboratory viral load testing, or care by an enrolled nurse and point-of-care viral load testing.⁹ After 12 months, people in the intervention arm were more likely to have been referred into a community ART delivery programme, and had 14% higher retention in care and viral suppression.¹⁰ However, the combined intervention was provided by research staff, meaning we do not know if point-of-care viral load assays can be implemented effectively in public sector clinics in low- and middle-income countries¹¹. Furthermore, STREAM only enrolled clinically stable patients (5% had viraemia at enrolment), and so there is inadequate data on the effect of point-of-care viral load testing amongst people with viraemia who are a vulnerable, priority population.”

Comment 3: Methods: The trial is presented as a feasibility study (line 116), yet the primary outcome is viral suppression at 24 weeks, which is more of an efficacy endpoint. Feasibility endpoints would be accrual and retention, etc. Because power calculations are based on viral suppression, I take this to be the primary endpoint, so this could perhaps be a pilot trial, but it is not a feasibility trial. The feasibility outcomes appear to be secondary outcomes. In addition, the aims of the qualitative study, to assess the feasibility of implementation within clinics will be hard to assess because there is no clinic-wide implementation as part of this trial, but random assignment within the clinic.

Response: Thank you for this comment. We recognise that the manuscript did not emphasise the exploratory nature of this study, and we now emphasise that we are not seeking to conduct a full trial to determine a definitive effect of point-of-care testing. Instead, this is a small study to guide the design of a future larger trial. We have used definitions of feasibility studies as described by Eldridge et al (full citation below), which describe pilot studies as ‘a subset of feasibility studies, rather than the two being mutually exclusive. A feasibility study asks whether something can be done, should we proceed with it, and if so, how. A pilot study asks the same questions but also has a specific design feature: in a pilot study a future study, or part of a future study, is conducted on a smaller scale.’

Given the above description, and that the future larger study could well be best designed as a cluster randomised trial (as commented by the reviewer below), we feel the present, individually randomised study is best described as a feasibility study, rather than a pilot study. To inform the design of the future study, we need a broad estimate of the effect of point-of-care viral load testing as well as estimates of accrual and retention, and insights into how best to implement the assay to manage HIV viraemia, hence our choice of outcomes. We designed the present feasibility study as an individually randomised study as this was the best way, with the resources available, to get an exploratory estimate of the effect of point-of-care testing compared to laboratory based testing.

Regarding the qualitative work, our approach is to introduce point-of-care testing into the clinic, but initially only for the subset of patients who are randomised at enrolment to later receive point of care testing as part of the trial. We will use qualitative work to evaluate this smaller scale implementation (i.e. setting up the assay in the clinic, ensuring testing occurs for the selected patients etc). If successful, findings will then be used to guide potential clinic wide implementation.

Therefore, we now highlight the exploratory nature of the estimate of the effect size of point-of-care testing:

Line 67: "POwER will provide new evidence to guide the development of interventions using point-of-care HIV viral load testing to improve the management of viraemia in low- and middle-income countries."

Line 72: "The study is limited by a moderate sample size and limited power to detect an effect of point-of-care viral load testing on viral re-suppression."

Line 115: "The overall aim of this feasibility study¹² is to provide exploratory estimates to guide the development of a future larger trial of point-of-care testing to manage HIV viraemia. Our specific objectives are to:

1. Broadly estimate the effect size of point-of-care viral load testing compared to standard laboratory viral load testing on viral re-suppression <50 copies/ml after 24 weeks.
2. Determine the feasibility of recruiting, randomising and following up patients in a randomised trial of point-of-care viral load testing to manage HIV viraemia.
3. Assess the perceptions of staff and the practical changes in clinic systems that are required to implement point-of-care viral load testing for a subset of patients in a primary care clinic in South Africa."

References: 12. Eldridge SM, Lancaster GA, Campbell MJ, et al. Defining Feasibility and Pilot Studies in Preparation for Randomised Controlled Trials: Development of a Conceptual Framework. PLoS one 2016;11(3):e0150205-e05. doi: 10.1371/journal.pone.0150205

Line 247: "We aim to broadly estimate the effect of point-of-care viral load testing using the primary outcome of viral suppression < 50 copies/mL at 24 weeks after enrolment, measured on a laboratory reference assay (e.g. Alinity m HIV-1) in both arms. Feasibility outcomes include proportions of participants enrolled, followed-up and who had same-day viral load testing in the point-of-care arm (Table 2)."

Line 264: "We will conduct exploratory analyses using Fisher's exact test to compare the proportions achieving binary study outcome measures in each arm, using an intention to treat analysis. As this is a feasibility study, these exploratory analyses will likely not be powered to test the hypotheses that there is a difference between the two arms for most outcome measures."

Line 348: "Results from this study will inform the design of future trials to assess the clinical effectiveness of point-of-care testing to manage viraemia."

Comment 4: The expected dates and rate of recruitment are not presented, nor a sample size in the methods, until lines 274-275 in which a period of 6 months is mentioned. This is a large omission and a highly non-standard way to conduct a RCT – usually the sample size is pre-determined, and back up plans are made if for some reason recruitment is lower than expected.

Response: Apologies for this omission, we have added the expected sample size to the abstract, and the beginning of the methods in the trial design section. In addition to the trial status section, we have added the date of first enrolment to the abstract.

Line 46: “We will enrol approximately 100 people living with HIV who are aged ≥ 18 years, receiving first-line ART but with recent viraemia ≥ 1000 copies/mL”

Line 130: “This will be a an open-label, single site, individually randomized, feasibility study of point-of-care viral load testing for the management of HIV viraemia (Figure 1). We aim to enrol approximately 100 adults receiving first-line ART and who have viraemia, and randomise them in a 1:1 ratio to receive point-of-care versus standard laboratory-based viral load testing. We will assess viral suppression after 24 weeks in each arm, and estimate recruitment, fidelity of the intervention and retention in care. We will also assess the implementation of point-of-care viral load testing using process evaluation data, in-depth interviews and focus group discussions.”

Line 296: “We conservatively estimate (based on our previous study)^{9,10} that we could enrol approximately 100 participants in 6 months. If enrolment proceeds better than anticipated, and resources allow us to enrol more, this will provide more precise estimates of the study outcomes. If enrolment proceeds slower than expected, we will consider expanding to another clinic site to achieve a minimum of 80 participants. Therefore, while we anticipate 100 participants, the final number may be between approximately 80-180 participants, and will be determined by the time and resources available.”

Comment 5: Table 3 notes that 240 might be eligible, and 100 are anticipated to be enrolled. Why so low? Also this should be discussed in the manuscript text, not just the table. It would be a severe study limitation. The number 240 should also be included in the figure.

Response: Thank you for noting this. We have amended this to clarify that 240 is the number of viral loads >1000 copies/ml that we estimate will occur during the study period at the clinic. Many of these will be in patients not eligible for the trial, as we describe below:

Line 634: “Assuming 240 viraemic patients during enrolment period, many of whom will not be eligible due to being on second-line ART, having had previous high viral loads, already having received enhanced adherence counselling or pregnancy.”

Comment 6: The motivation for the feasibility outcomes to inform a future larger trial is questionable, given that larger trials (if really needed or ethical, see above), should be conducted as cluster RCTs so that clinic level barriers/facilitators, which are may drive efficacy more than patient variability, can be examined. There is no justification for feasibility outcomes (a and b) for this type of trial – many trials have been conducted in public clinics and the extra anticipated barriers for this particular trial are not described. The outcomes of the point of care testing (i.e. same day results) are justified.

Response: As stated above, we do not agree that the one, single-site clinical trial which we conducted in people largely stable on ART is enough to justify mass rollout of point-of-care viral load testing; further evidence in different clinical groups and from larger, multi-site randomised controlled trials is needed. In particular, we feel that people with viraemia are the most likely to benefit and therefore a trial targeting this group for point-of-care testing is justified. There are no other studies, planned or underway, assessing this that we are aware of. We agree that a cluster RCT design may be

appropriate for the future trial, but an individually randomised trial may also be appropriate, and either way, outcomes from this feasibility study will be useful to inform the choice of design. Specifically, we can clarify that outcome 2b) (follow-up) will be estimated in each arm (in case point-of-care testing influences retention in care and study follow up). We agree that outcome 2a) (enrolment) is less unique to point-of-care trials, and therefore could be estimated from other studies, but as we are running this study, we feel it would be remiss not to measure this. We agree that outcome 2c) (fidelity of point-of-care testing) is crucial.

Line 280: "To assess study follow-up, we will calculate the proportion of participants who attend the study exit visit at 24 weeks after enrolment, overall and in each arm in case point-of-care testing influences retention."

Table 2: "b) Proportion of enrolled participants who attend the study exit visit overall and in each arm"

Comment 7: Discussion: The discussion does not address some serious limitations of the study. There is no discussion of whether a 90-minute wait is really point of care. There is no discussion of the limitations of a single site trial. There is no discussion of the ramifications of the site chosen for the trial -- a large clinic – and whether it would be representative of a large sector of care. There is no discussion of the limitations of individual randomization versus a cluster trial design.

Response: Thank you for these suggestions which we agree should have been discussed. We have included them in the limitations section as below:

Line 391: "In HIV programmes, some tests such as rapid, lateral flow assays for diagnosis of HIV, have been evaluated, endorsed and incorporated into WHO guidelines and successfully adopted in many settings.³⁸ However, other assays, such as more complex molecular polymerase chain reaction technologies for tuberculosis, have remained as laboratory tests despite being marketed as point-of-care assays.³⁷ Given that the Xpert HIV-1 VL and m-PIMA assays use similar platforms and take over 1 hour for results to be available, they have also been criticized as not being implementable as true point-of-care tests."

Line 419: "Furthermore, findings from this large urban, clinic may not be applicable to other settings, including smaller clinics in rural settings where advantages of point-of-care testing over laboratory-based testing may differ."

Line 421: "Our future trial may be best designed as a cluster-randomised trial, with individual clinics randomised to receive the point-of-care testing intervention. While POWER is an individually randomized study, estimates of the effect of point-of-care testing will remain valuable for design of potential cluster randomized or individually randomized future studies."

Comment 8: The study design with uncertain sample size is a huge limitation, and there is no discussion of this. The justification that this is a feasibility study does not make sense – even feasibility studies have set sample sizes, and as noted above, the primary outcome is not a feasibility outcome.

Response: As stated above, we agree that we had not made it clear enough in the original submission that this is intended as a feasibility study, assessing exploratory estimates of effectiveness of point-of-care viral load testing, and with a small target sample size (100 participants) that is not powered to detect a difference between the two arms. We have therefore deleted the power calculation assuming a sample size of 180, as well as other changes in response to the reviewer's comments above. We hope that our changes now make this clear. We do not see the methodological issue with enrolling more patients if enrolment proceeds better than anticipated, as

this will provide more precise estimates of study outcomes. This has been approved by both relevant ethics committees/the study sponsor.

VERSION 2 – REVIEW

REVIEWER	Nancy Puttkammer University of Washington, Seattle WA USA One of the co-authors, Dr. Paul K. Drain, is a colleague in the Department of Global Health at University of Washington. I have no previous or current collaborative work with Dr. Drain.
REVIEW RETURNED	01-Feb-2021
GENERAL COMMENTS	Thank you for addressing the reviewer comments in a complete manner.